# Trends and Gaps in the Scientific Literature about the Effects of Nutritional Supplements on Canine Leishmaniosis

**DOI:** 10.3390/pathogens13100901

**Published:** 2024-10-14

**Authors:** Annalisa Previti, Vito Biondi, Diego Antonio Sicuso, Michela Pugliese, Annamaria Passantino

**Affiliations:** Department of Veterinary Sciences, University of Messina, 98168 Messina, Italy; annalisa.previti1@unime.it (A.P.); vito.biondi@unime.it (V.B.); diego150899@gmail.com (D.A.S.); passanna@unime.it (A.P.)

**Keywords:** canine leishmaniosis, machine learning, nutraceutical, nutritional supports, text mining, topic analysis

## Abstract

In canine leishmaniosis (CanL), complex interactions between the parasites and the immunological background of the host influence the clinical presentation and evolution of infection and disease. Therefore, the potential use of nutraceuticals as immunomodulatory agents becomes of considerable interest. Some biological principles, mainly derived from plants and referred to as plant-derived nutraceuticals, are considered as supplementation for *Leishmania* spp. infection. This study provides a systematic review regarding the use of nutraceuticals as a treatment using a text mining (TM) and topic analysis (TA) approach to identify dominant topics of nutritional supplements in leishmaniosis-based research, summarize the temporal trend in topics, interpret the evolution within the last century and highlight any possible research gaps. Scopus^®^ database was screened to select 18 records. Findings revealed an increasing trend in research records since 1994. TM identified terms with the highest weighted frequency and TA highlighted the main research areas, namely “Nutraceutical supports and their anti-inflammatory/antioxidant properties”, “AHCC and nucleotides in CanL”, “Vit. D3 and Leishmaniosis”, “Functional food effects and Leishmaniosis” and “Extract effects and Leishmaniosis”. Despite the existing academic interest, there are only a few studies on this issue so far, which reveals a gap in the literature that should be filled.

## 1. Introduction

Canine leishmaniosis (CanL) is one of the most relevant zoonotic diseases affecting dogs and humans in many parts of the world. It is endemic in the region of the Mediterranean basin, Africa, Asia, and South America [1]. This disease is caused by parasites of the genus *Leishmania* (*Kinetoplastida: Trypanosomatidae*), which are primarily transmitted by phlebotomine sandflies (*Diptera: Psychodidae: Phlebotominae*) [2]. Globally, *Leishmania infantum* is the most common species involved in the etiology of CanL [1]. The most common species involved worldwide in the etiology of CanL is *L. infantum*, while infections from *L. major* and *L. tropica* are considered sporadic [1]. This protozoan is transmitted from one host to another by the bite of infected female phlebotomine sand flies. The dog represents the main natural reservoir of the parasite, playing a crucial role in maintaining the cycle of transmission to dogs and humans [2]. The host can manifest different forms due to the severity and location of lesions [3]. The spectrum of clinical and pathological findings is wide, ranging from subclinical and self-limiting to severe disease, and related to the predominant response of the immune system. The immune response plays an important role in the progression and outcome of diseases [4,5]. In humans, the disease may occur in different forms: (i) an asymptomatic form confined to the skin (Cutaneous leishmaniasis, CL), with rare lesions, which is the most common form, accounting for approximately 95% of cases in America, the Middle East, the Mediterranean basin and Central Asia, with an annual estimate of 600,000 to 1 million new cases [6]; (ii) Mucocutaneous leishmaniasis (MCL), which is primarily reported in Bolivia, Ethiopia, Brazil, and Peru can be fatal if left untreated [6]; and (iii) visceral leishmaniasis (VL), also known as kala-azar, with an estimated 50,000 to 90,000 new cases annually worldwide [6], affects the entire reticule-histiocytic system, spleen, and internal lymph nodes, and, likewise to the MCL form, VL is fatal if left untreated [7]. Unlike, infected dogs can remain clinically healthy or develop clinical signs in relation to different variables such as the specific genetic background, proper pathogenic mechanisms of the parasite and the variable immune response of each host [8]. Treatment of dogs for *Leishmania* spp. infection is aimed at reducing the parasitic load. This depends on increasing intracellular oxidant compounds for the killing of *L. infantum*. However, an excess of oxidative species and the depletion of antioxidants can lead to oxidative stress, resulting in increased and widespread inflammation [9]. Pentavalent antimonials are commonly used to treat CanL, either alone or in combination with allopurinol. While effective against the disease, these drugs have drawbacks such as high cost, unwanted side effects from injections, and the potential developing resistance [10]. Amphotericin B is not widely used because it requires close renal monitoring, and its effectiveness is not entirely clear [11,12,13,14]. Pentamidine isethionate is also rarely used due to its toxicity [15]. Although aminosidine (paromomycin) may have good clinical efficacy, its use is not recommended, mainly due to its potential nephrotoxicity and ototoxicity [16]. Other oral treatments like miltefosine have also shown resistance and numerous adverse effects, including teratogenicity and gastrointestinal issues [17,18]. This has led to a search for safer treatment options. In this context, nutraceutical substances play an important role in counteracting the side effects of specific *Leishmania* spp. treatment options and modulating the host’s immune defense [19]. For example, Eddin et al. (2022) [19] reported that α-Bisabolol, derived naturally from essential oils of many edible and ornamental plants, can be used as a dietary agent, a nutraceutical or phytopharmaceutical agent or as an adjuvant based on currently available modern medicines. De Sousa Gonçalves et al. (2021) [9] reported that the addition of nutritional adjuvants (NAs) like omega-3, B vitamins and polyunsaturated fatty acids to anti-*Leishmania* drugs (ALDs) in the treatment of CanL would be clinically beneficial. According to Dea-Ayuela et al. (2020) [10], dietary nucleotides plus Active Hexose Correlated Compound (AHCC) have shown helpful effects in dogs with clinical leishmaniosis and clinically healthy *Leishmania*-infected dogs; that review has evaluated nucleotides and AHCC potential leishmanicidal activity by quantifying nitric oxide (NO) production and replication of parasites with encouraging results. All these substances improve patients’ conditions only if used in specific diets as an integration of common drug treatments.

The present study provides a systematic review relating to the use of nutraceuticals as a new drug treatment using a text mining (TM) and topic analysis (TA) approach, identifying dominant topics of nutritional supplements in leishmaniosis-based research, summarizing the temporal trend in topics, interpreting the evolution of topics within the last century and highlight the possible research gaps.

## 2. Materials and Methods

### 2.1. Literature Search and Descriptive Statistics

A systematic scientific literature search was carried out to identify papers with at least an English abstract that covered the topic of the effects of nutritional supplements in *Leishmania* spp. infection, using Scopus^®^ (i.e., the abstracts and citation database of Elsevier©). The Scopus^®^ database was selected due to its accessibility and comprehensive coverage of the academic literature [20]. The search was conducted on the 1 August 2024, and it was refined based on article type (review and scientific article), language (English) and availability of abstract. The keywords used included the following: “nutritional support AND leishmaniosis”, “nutraceutic AND leishmaniosis”, “nutritional AND supplement AND leishmaniosis”, “nutraceutical AND leishmaniosis”, and “immunonutrient AND leishmania”. A Microsoft Office Excel^®^ (version n.16, Microsoft Office 365, Microsoft Corporation, Washington, DC, USA) spreadsheet containing all records published was retrieved from Scopus^®^; in the database, each line reported a document and each column its information, namely authors, affiliations, abstracts, year of publication, type of document (e.g., article or review), source of publication (e.g., journal title) and topic. The records were then screened and those that had no abstract, no articles/reviews (book chapters, conference papers, erratum, letters, notes, short reviews), no author name, no English text (i.e., articles in German) and/or duplicates were excluded automatically. Finally, manual screening was performed by the authors (Apr, AP, VB and MP) to decide the eligibility of the record for inclusion in the final analysis. Records related to other aspects of *Leishmania* spp. infections diagnosis and treatment were excluded. The flowchart (Figure 1) illustrates each step of the process, showing the number of records kept for further analysis or removed from consideration.

A total of 178 records were identified. Preliminarily, descriptive statistics were conducted to create a detailed profile of the scientific dataset, focusing on the year and journal of publication. Pivot tables were employed to determine the yearly document count and highlight the leading journals that play a significant role in the issue.

### 2.2. Text Mining

After the selection of articles for analysis, Rstudio was utilized for text mining in R (Version 1.3.1093, Free Software Foundation, Boston, MA, USA) following the download process. A dedicated sheet was prepared with two distinct columns: “doc_id”—containing the sequential numbering of the 18 documents—and “text”, which encompassed the abstracts of the chosen papers for text mining (TM) analysis. The corpus of documents was submitted to pre-processing steps, as reported by Sebastiani (2002) [21]. Specifically, all characters were converted to lowercase; unusual symbols (as well as “@”, “/” or “*”), punctuations, numbers and stop words (such as “the”, “a”, “and”, “on”, etc.) were removed. The authors also removed words strictly relating to the researched topic or that are commonly used (such as “leishmaniosis”, “leishmania”, “nutraceutic”, “support”, “supplements”, “differ”, “level”, “group” and “diet”). Extra whitespaces that occurred in previous steps have also been excluded. To reduce to their root forms, the text has been tokenized. Afterward, a document-term matrix (DTM) was constructed, aligning documents along the rows and terms along the columns. To assign relative weights to words, a term frequency-inverse document frequency (TF-IDF) technique has been applied considering both their frequency within a document and their prevalence in the document collection. This adjustment improved the evaluation of the meaning of a word within the document set, with the relevant words (TF-IDF > 0.15) visualized in a histogram. Also, a word cloud representing the most relevant words was created using the following website https://www.wordclouds.com/ (accessed on 10 August 2024), where larger character sizes indicated higher TF-IDF values. Associations between the most relevant words (TF-IDF > 0.15) and all document terms in the corpus were identified based on a correlation threshold of ≥0.2. R packages (2017) and functions from “tm”, “SaeballC”, “ggplot2”, “delve” and “tidyverse” were used for the statistical analysis.

### 2.3. Topic Analysis

The Latent Dirichlet Allocation (LDA) method was applied to perform TA. The LDA is a hierarchical Bayesian approach that identifies thematic topics by analyzing the co-occurrence of words in texts [22]. Each topic is represented as a distribution of words, and each text as a distribution of latent topics. By analyzing the observed texts and words, the model uncovers the underlying topic structure, generating topic distributions for each text and word distributions for each topic [23]. The LDA function with Gibbs sampling from the “topic models” package in R (Version 1.3.1093, Free Software Foundation, Boston, MA, USA) was employed. The most common words for each topic and their relative probabilities were visualized using the “tidytext” R library. Before beginning the analysis, the number of topics into which to divide the corpus had to be determined. Since the optimal number of topics is generally unknown, it has been experimented with 4, 5, 6 and 7 topics, selecting the most informative set based on consensus. After settling on five topics, the research group named them with indicative labels. To classify the topics, the cumulative probabilities of the top words in each topic were then calculated, and the topics were presented based on this ranking. Each topic was depicted in a bar histogram, with each bar representing the probability of a word within that topic (measured by beta-value coefficient). This visualization method, according to Nalon et al. (2021), assigned a name to each topic for easier identification [24].

## 3. Results

### 3.1. Descriptive Statistics

Out of 178 abstracts downloaded by Scopus, 18 (10.11%) fulfilled the screening and eligibility criteria and were retained. Articles about other aspects of CanL (46.6%; *n* = 83) were excluded. The reasons for exclusions were the presence of duplicates (16.3%; *n* = 29), the absence of abstract (6.7%; *n* = 12), no English full text (0.56%; *n* = 1) and no articles and/or reviews (19.7%; *n* = 35). The type of records retained were research articles (*n*.14/18; 77.8%) and reviews (*n*.4/18; 22.2%). The total number of records published per year has increased in the last decade (Figure 2).

The records were published in 15 different scientific journals of which those with more than one article on the topic are “Veterinary Parasitology” (with 2/18 records; 11.1%) and “Animals” with 3/18 records (16.7%) (Figure 3).

Table 1 shows the most cited documents, specifically the title of the publication, and the citations are shown for each article.

The first most cited publication in the last 10 years was “Functional foods in pet nutrition: Focus on dogs and cats” [25], with 65 citations, which evaluated the importance and the potential of functional foods in pet nutrition, focusing on dogs and cats. Regarding CanL, it is suggested that a specific diet may determine a modulator effect on the immune response in association with traditional pharmacological management. Eddin et al. (2022) [19] was the second most cited in ex aequo with Rondon et al. (2011) [26] with 39 citations. Eddin et al. [19] focused on the potential biocidal effects of α-bisabolol on *L. infantum* promastigotes. This substance obtained from *Matricaria chamomilla* induces apoptosis, resulting in the externalization of phosphatidylserine and disruption of the plasma membrane. It also increases the secretion of IFNγ and triggers a Th1 cell-mediated immune response. Furthermore, it causes a reduction in the mitochondrial membrane potential and ATP levels. Rondon et al. (2011) [26] assessed in vitro the impact of fractions derived from *Aloe vera* (aloe), *Coriandrum sativum* (coriander) and *Ricinus communis* (castor) on promastigotes and amastigotes of *L. infantum* to analyze their cito toxicity into murine monocytic cells. The article of Cortese et al. (2015) [27] was the third most cited, with *n*.30 citations focusing on the inclusion of nutraceuticals in the nutritional regimen of dogs affected with CanL. It reports a positive correlation with the reduction in Th1 cells and an increase in Treg cells, suggesting that the administration of a targeted dietary supplement may enhance the clinical response to the conventional treatment regimen in CanL.

Of the articles included in the analysis, those published by Segarra et al. (2018) [28] were referenced the fourth most often, receiving *n*.24 citations. It provides a comprehensive review of the scientific evidence on the immunomodulatory effects of dietary nucleotides across a range of animal species. Also, it presents a CanL management algorithm that incorporates nucleotides. The last most frequently referenced article, authored by Da Silva Bezerra et al. (2019) [29], assesses the impact of dietary vitamin D on the progression of *L. amazonensis* infection in mice.

### 3.2. Text Mining

Following data pre-processing and sparseness reduction a total of *n*.852 terms were preserved from the initial *n*.18 records. Histograms included in Figure 4 show the most significant words (TF-IDF ≥ 0.15) based on the TF-IDF weighting system (Figure 4).

Additionally, Figure 5 illustrates a word cloud where the font size corresponds to the TF-IDF value of each word. The words with the highest TFIDF were “extract”, “function”, “nucleotid”, “mice”, “ahcc”, “promastigot”, “plant”, “cell”, “serum”, “fraction”, “αbisabolol”, “nutraceut”, “oxid”, “observ”, “amastigot”, “avail”, “clinic”, “concentr”, “effect”, “acid”, “respon”, “mgml”, “pharmacolog”, “level”, “decrea”, “patien”, “quercetorum”, “administr”, “vitd”, “dietary”, “treg” and “infect” (Figure 5).

A correlation coefficient of ≥0.4 was found between the most frequent words with a TF-IDF score of 0.15 or higher (Table 2).

### 3.3. Topic Analysis

Five topics were chosen as the ideal topics and labels were assigned to each of them. The name of each topic as well as the number of records contained in each topic and their first-year publication are shown in Table 3.

Figure 6 shows the topics numbered from 1 to 5 according to the cumulative probabilities (cp), and the first 10 words for each topic numbered from 1 to 5 according to the cumulative probabilities (cp), namely “Nutraceutical supports and their anti-inflammatory/antioxidant properties”, “AHCC and nucleotides in CL”, “Vit. D3 and Leishmaniosis”, “Functional food effects and Leishmaniosis” and “Extract effects and Leishmaniosis”, represented by 5, 3, 3, 2 and 5 documents, respectively (Figure 6).

## 4. Discussion

This study explored the effects of nutraceutical supports in CanL using sophisticated machine learning techniques, like TM and TA. By analyzing a wide range of scientific literature, the authors were able to delve into the complexity of this matter. Through these methodologies, they were also able to pinpoint areas of limited understanding and gaps in knowledge.

The number of published articles on the effects of nutritional supplements in CanL has demonstrated a gradual increase since 2011, reaching a notable peak in 2022. This trend is not unexpected; nevertheless, it has long been recognized that plant-derived compounds possess therapeutic value [30]. Indeed, in recent years, nutraceuticals have gained considerable popularity in various medical fields, largely due to their recognized physiological effects and potential for promoting health [31]. There is no official definition for nutraceuticals. The term nutraceutical is used to describe multi-target substances used in low concentrations. These compounds are cultivated, manufactured or extracted and, when administered to individuals, are capable of improving their health and well-being, as opposed to “functional foods”, which are considered foods fortified with vitamins, proteins and carbohydrates [31]. Their effects include anti-inflammatory, anti-cancer, antioxidant and prebiotic properties, as well as the regulation of lipid metabolism [31]. Furthermore, within the field of veterinary medicine, there has been a discernible trend towards pet owners displaying a heightened concern for the health and overall welfare of their animal companions [32]. This shift in attitude is accompanied by growing knowledge and awareness of conditions such as leishmaniosis, prompting a surge in interest in exploring the use of modern pharmacological tools for the treatment and control of such diseases [33,34,35]. As pet owners become more educated about the various health issues that can affect their beloved pets, there is a corresponding increase in the demand for advanced and effective treatment options that can provide optimal care and management of these conditions. This evolving landscape underscores the importance of leveraging the latest developments in pharmacology to address the healthcare needs of animals and meet the growing expectations of pet owners seeking the best possible outcomes for their animals [33,34,35].

The first words that appear, in order of importance and closeness of meaning, may indicate that the most studied aspects of dietary supplementation in CanL are related to “plant extracts”, “nucleotides”, “active hexocorrelated compounds” (AHCC), “alpha-bisabolol”, “quercetorum” and “vitamin D”. It is worth noting that the word “extracts” occurs more frequently in TM analysis. Among these, i.e., *Olea europaea* subsp. *laperrinei* extracts show the existence of numerous anti-leishmanial biomolecules such as oleuropein, hydroxytyrosol, rutin, gallic acid, caffeic acid, rosmarinic acid and quercetin [3]. In addition, the interactions of four major metabolites of *P. quercetorum*, including α-pinene, α-fenchyl acetate, limonene and trans-β-caryophyllene, with the main virulence factor of *L. major* were investigated in the literature [36]. Surprisingly, the in vitro results indicated that the extract of *P. quercetorum* had a cytotoxic effect on amastigotes and promastigotes of *L. major* in a non-dose-dependent manner [37]. It is worth noting that the words “promastigote” and “amastigote” also appear more frequently in TM, probably because these are the main targets of these extracts.

Moreover, *Capparis spinosa* L., *Ricinus communis* and *Solanum luteum* were used as lethal agents for the promastigotes of *Leishmania major* [38].

A further consideration can be made based on the associations between the words as follows. For example, the word “concentr” is often associated with “extract” and “effect”, probably due to the correlation between the dose and effect of the extract fraction [36]. However, as noted previously, some, such as *P. quercetorum*, had a cytotoxic effect on *Leishmania* spp. that was not dose-related [39]. In addition, “serum” is associated with “Treg”, “Vitd” and “α-bisabolol”, because the serum level of these substances correlates with its T-regulatory capacity [19,39].

This analysis highlighted the most important research focused on nutritional support in dogs. There is a strong correlation between these topics and their interrelationship is readily apparent. The main topics of interest are “Nutraceutical supports and their anti-inflammatory/antioxidant properties” and “Extract effects and leishmaniosis”, which are the topics with the most and the oldest papers (2015 and 2011, respectively), followed by topic 4 (“Functional food effects and Leishmaniosis”), which is strictly connected in its meaning.

Indeed, in CanL, the presence of oxidative species within the host has been identified as a sign of oxidative stress [9]. Some authors have highlighted the potential of immunonutrition in the management of inflammatory conditions by mitigating and averting tissue damage caused by oxidative stress [9]. Previous studies have demonstrated that antioxidant nutraceuticals have a protective effect on cell membranes, resulting in less tissue cell death and less inflammation due to homeostatic debris removal processes in inflammatory diseases [40,41]. Omega-3 polyunsaturated fatty acids (PUFAs) are among the common excipients known for their antioxidative properties and anti-inflammatory effects, as extensively documented in the literature [9]. The incorporation of anti-leishmanial drugs, along with early administration of omega-3 PUFA and B vitamins, has been found to facilitate prompt clinical improvements, indicating restoration of renal function and reduction in lipid and protein peroxidation levels [9].

The utilization of phytonutraceuticals has shown promise in regulating immune response and the integration of nutraceutical pet food with conventional treatment can modulate the immune response in CanL [27]. Nutraceutical supplementation could be associated with modulation of the Th1 inflammatory response, which would preserve tissues from immune-mediated damage. Furthermore, immunomodulation induced by nutraceuticals can also improve the overall clinical picture in animals [42]. Additionally, a blend of Krill oil, dry mushrooms (*Cordyceps sinensis*), Krill flour, Gentian (*Gentiana lutea* L.) dry root and herb-derived products (*Eleutherococcus senticosus* L.) has been developed to bolster the natural physiological defenses of dogs and cats exposed to external aggressions and has demonstrated effectiveness in dogs with leishmaniosis, showing improvements in inflammatory and oxidative markers, as well as the amelioration of clinical signs [8]. The antileishmanial activity of medicinal plants, such as *Artemisia annua*, *Plumbago scandens*, *Physalis angulata*, *Phyllanthus amarus*, *Piper aduncum*, *Peschiera (Tabernaemontana) australis* and *Kalanchoe pinnata*, is an interesting way to treat the skin leishmaniosis symptoms [43]. Finally, an innovative perspective on the pharmaceutical use of Sargassaceae algae by-products in the fight against leishmaniosis was reported by Abdoul-Latif F.M. et al., 2024 [43].

Additional important areas of focus in the treatment of leishmaniosis with nutritional supplements include nucleotides and active hexose correlated compounds (AHCC). Unsurprisingly, this is a single topic of interest, and “nucleotide” and “ahcc” appear as the most common words in TM. Nucleotides play an important role in immune competence, intestinal development and recovery. Nucleotides are essential low-molecular-weight biological molecules, and exogenous sources of them become indispensable in circumstances characterized by stress and heightened demand for nucleic acid synthesis, such as periods of immunosuppression, infection and certain disease states, due to their capacity to modulate immune responses, to positively impact lipid metabolism, immunity, as well as tissue growth, development and repair [44]. On the other hand, AHCC, a cultured extract derived from shiitake mycelium, has been used in human health for immunostimulating benefits, particularly in boosting cellular immunity [45]. A study in which nucleotides and AHCC nutritional supplement were administered orally to clinically healthy L. infantum-infected dogs for 365 days showed that it was safe and resulted in a significant reduction in ELISA serological titers of antibodies against *Leishmania* spp., as well as a reduction in the rate of disease progression [28]. These results highlight the preventive efficacy of the nutritional supplements in CanL, indicating a reduced likelihood of progression to diseased states compared to non-supplemented dogs [28].

Vitamin D treatment has been identified as having a significant impact on *L. mexicana* infection, demonstrating a protective effect by limiting lesion development in mice [46]. In the context of canine infection caused by *L. infantum*, Vitamin D also plays a beneficial role, with the progression of the infection being closely linked to Vitamin D deficiency in dogs [47]. Conversely, knockout mice exhibit enhanced resistance to *L. major* infection [47], indicating a potentially detrimental role for Vitamin D in this particular scenario. It appears that the influence of Vitamin D on *Leishmania* spp. infection outcomes vary depending on the specific species of the parasite involved.

It is worth mentioning that certain nutraceuticals, despite lacking robust scientific backing, have been utilized by veterinary professionals, potentially due to their inclination towards these substances [9,48]. One example of this is Corpet^®^ (Mycology Research Laboratories Ltd., Luton, Bedfordshire, UK), a mushroom-based (Coriolus versicolor) nutritional product that has been employed for the extended treatment of CanL [9]. Assessing their effectiveness would require further research.

## 5. Conclusions

This review utilized machine learning methodologies to investigate the literature about to effects of nutritional supplements in CanL. The analysis revealed a notable increase in interest in dietary supplements for CanL, reflecting a growing emphasis on this topic within the scientific community. However, it is necessary to acknowledge the methodological limitations inherent in this review. The chosen search strings may have inadvertently excluded potential synonyms, potentially constraining the scope of the study. Additionally, the decision to confine the search to records solely within the Scopus^®^ database may have led to the inadvertent exclusion of pertinent sources of information, thereby influencing the overall comprehensiveness and depth of the review. The specific search parameters employed during the review process may have further narrowed the pool of records available for comprehensive analysis. Notwithstanding these inherent limitations, this review managed to provide valuable insights and critical observations on the subject matter. Through a meticulous examination and synthesis of the findings from eighteen selected papers, the review provided valuable insights into the matter, and identified persistent knowledge gaps, offering guidance for future research endeavors in this domain.

In conclusion, while acknowledging the methodological constraints, this review on nutritional supplements in CanL represents a significant contribution to the existing literature, offering valuable insights and paving the way for future research initiatives about this topic.

## Figures and Tables

**Figure 1 pathogens-13-00901-f001:**
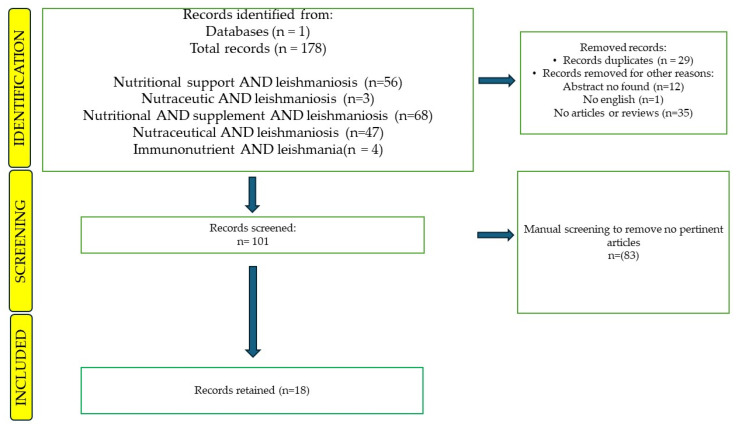
The flowchart illustrates each step of the process regarding the selection and inclusion of articles.

**Figure 2 pathogens-13-00901-f002:**
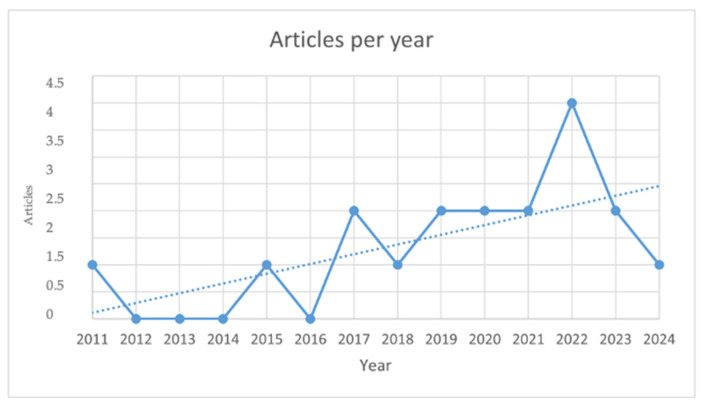
The total number of records published per year about the topic considered are illustrated in this figure.

**Figure 3 pathogens-13-00901-f003:**
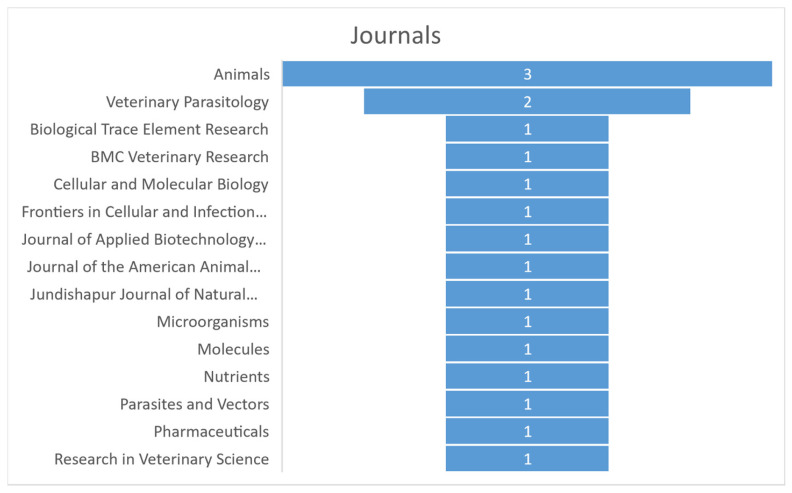
The most representative scientific journals publishing articles regarding the topic.

**Figure 4 pathogens-13-00901-f004:**
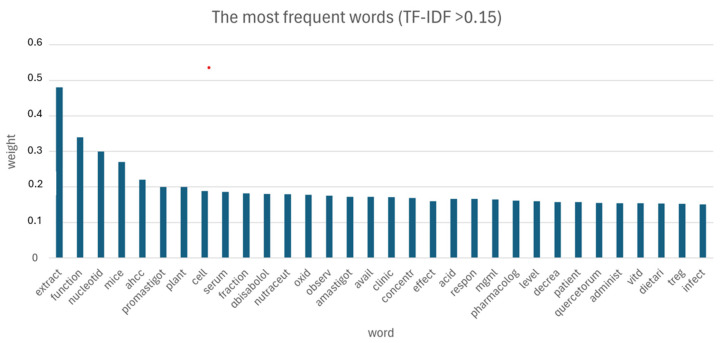
Histograms show the most significant words (TF-IDF ≥ 0.15) based on the TF-IDF weighting system.

**Figure 5 pathogens-13-00901-f005:**
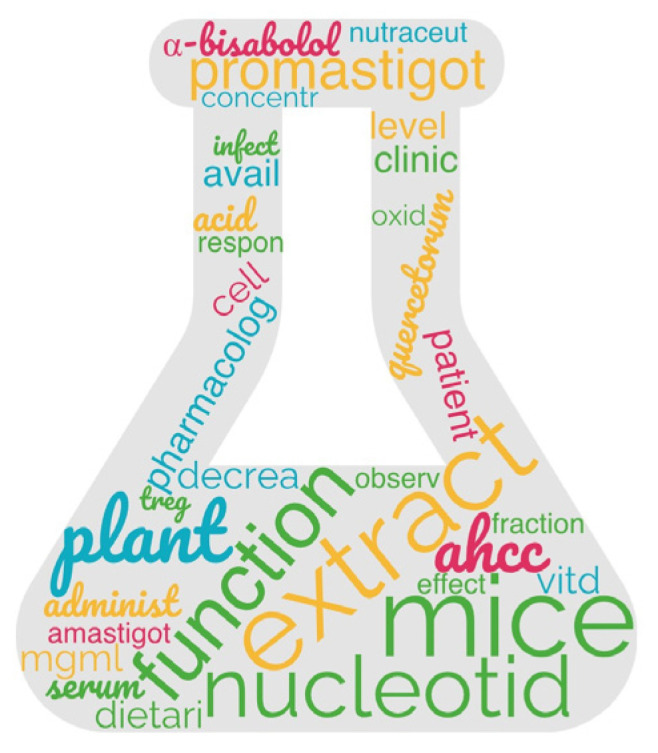
The words’ cloud includes the different words with the highest TFIDF. The font size corresponds to the TF-IDF value of each word.

**Figure 6 pathogens-13-00901-f006:**
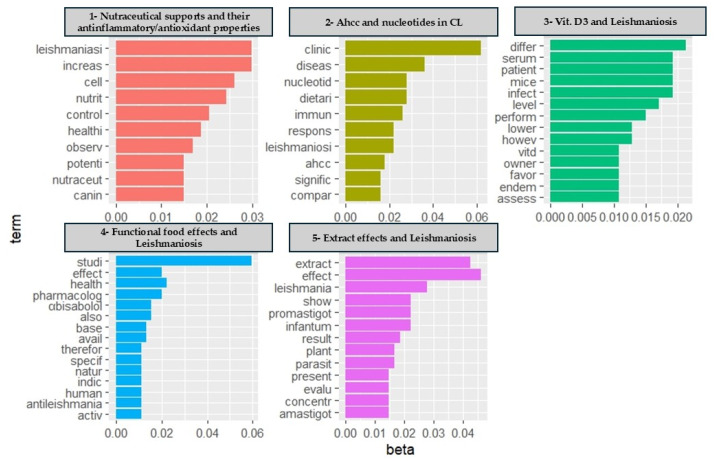
Topics are numbered from 1 to 5 according to the cumulative probabilities (cp), and the first then words for each topic are also numbered. Histograms report the most frequent 15 words (terms) for the 5 topics identified from the scientific literature analysis. The “beta” shows the relative probability of each word in each topic.

**Table 1 pathogens-13-00901-t001:** The most cited documents. The authors, the year of publication, the journal and the title of publication are detailed in the column.

No.	Authors/Year/Journal	Title of the Publication	JC
1	Di Cerbo A. et al.; 2017; Research in Veterinary Science [25]	Functional foods in pet nutrition: Focus on dogs and cats	65
2	Eddin L.B. et al., 2022; Nutrients [19]	Health benefits, pharmacological effects, molecular mechanisms, and therapeutic potential of α-Bisabolol	39
3	Rondon F.C.M. et al., 2011; Advances in Skin and Wound Care [26]	In vitro effect of Aloe vera, Coriandrum sativum and Ricinus communis fractions on *Leishmania infantum* and on murine monocytic cells	39
4	Cortese L. et al., 2015; BMC Veterinary Research [27]	An immune-modulating diet increases the regulatory T cells and reduces T helper 1 inflammatory response in leishmaniosis affected dogs treated with standard therapy	30
5	Segarra S. et al., 2018; Parasites and Vectors [28]	Prevention of disease progression in *Leishmania infantum*-infected dogs with dietary nucleotides and active hexose correlated compound	24
6	Da Silva Bezerra I.P. et al., 2019; Frontiers in Cellular and Infection Microbiology [29]	Dietary vitamin D3 deficiency increases resistance to *Leishmania (Leishmania) amazonensis* infection in mice	10

JC: journal citations.

**Table 2 pathogens-13-00901-t002:** Associations between the most relevant words (with TFIDF ≥ 0.15).

Words (TF-IDF ≥ 0.15)	Association between Most Frequent Words (Correlation Grade ≥ 0.4)
Acid	administ, ahcc, amastigot, avail, cell, clinic
Concentr	decrea, dietari, effect, extract, fraction, function
Infect	level, mgml, mice, nucleotid, nutraceut, observ
Oxid	patient, pharmacolog, plant, promastigot, quercetorum, respon
Serum	treg, vitd, α-bisabolol

**Table 3 pathogens-13-00901-t003:** Different topics and the number of records contained within the first year of publication.

Topic Number	Label of the Topic	Papers (*n*)/from Year
1	Nutraceutical supports and them Anti-inflammatory/antioxidant properties	5/2015
2	AHCC and nucleotides in CL	3/2018
3	Vit. D3 and leishmaniosis	3/2019
4	Functional food effects and leishmaniosis	4/2017
5	Extract effects and leishmaniosis	5/2011

*n*: number.

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
