# Peer review of "Trends and Gaps in the Scientific Literature about the Effects of Nutritional Supplements on Canine Leishmaniosis"

_pathogens, 2024, doi:10.3390/pathogens13100901_

Round 1
Reviewer 1 Report
Comments and Suggestions for Authors
Canine leishmaniasis is a parasitic infection caused by trypanosomatid protozoa, parasites. It is a disease (zoonosis) that can be transmitted from animals to humans and vice versa, with the insect vector being known as the sand fly. In fact, it is a serious zoonosis that can lead to the death of both humans and infected animals. Therefore, this disease is a public health issue that requires everyone to take care in its prevention and control.
When in contact with its host (in this case, the dog), the Leishmania parasite begins to attack the phagocytic cells. When the leishmania is captured by the phagocytes, it begins to differentiate into a form known as amastigote, this evolutionary form has the capacity to proliferate inside the host cells. The infection can reach organs such as the liver, spleen and bone marrow.
There are two types of leishmaniasis: cutaneous and visceral. Cutaneous leishmaniasis is caused by two types of parasites: Leishmania braziliensis and Leishmania mexicana. Visceral leishmaniasis is caused by the parasites Leishmania donovani, Infantum and Chagasi. However, it is important to know that in 99.9% of cases when the topic is leishmaniasis in dogs, it is canine visceral leishmaniasis that is being discussed. This is because cutaneous leishmaniasis does not target dogs, whereas visceral leishmaniasis does.
In this interesting review, Annalisa Previti and colleagues provided an interesting systematic review related to the use of nutraceuticals as a novel drug treatment using a text mining (TM) and topic analysis (TA) approach. This retrospective study aims to identify dominant topics of nutritional supplements in leishmaniasis-based research, summarize the temporal trend of the topics, interpret the evolution of the topics over the last century and highlight any possible research gaps.
The review is interesting, readers will find a pleasant text and the topic is of interest to different areas of science (immunology, parasitology, medicine, veterinary medicine).
Author Response
Dear Reviewer,
Thank you very much for your time and all your comments.
Reviewer 2 Report
Comments and Suggestions for Authors
Line 9 – replace “the parasite” with “the parasites”
Line 10 – immunological BACKGROUND
Adapt to read as: In canine leishmaniosis (CanL) there are complex interactions between the parasite and the immunological backgrounds of the host influencing the clinical presentation and evolution of infection and disease. Therefore, therefore the potential use of nutraceuticals as immunomodulatory agents becomes of considerable interest.
Line 12 – please refine the definition of nutraceuticals
Line 20 – adapt: AHCC and nucleotides in CanL
Keywords – please display alphabetically
Line 27 – Canine leishmaniosis (CanL)
Line 29 – a flagellated protozoan
Line 31 – sand flies
Line 33 – the disease is not transmitted, the agent is – adapt text
Line 34 – replace localization with location
Line 38 – human cutaneous leishmaniosis should not be considered an asymptomatic form – please correct text
Line 41 – mucocutaneous leishmaniasis
Line 42 – Peru (not Perù)
Line 43 – visceral leishmaniasis [without quotation marks]
Remark – please select leishmaniosis or leishmaniasis – and use only one of the words, because they are synonymous – change accordingly throughout the manuscript
Line 45 – start a new sentence: Unlike…
Line 45 – MCL can also be fatal
Line 46 – replace asymptomatic [applicable only to people] with clinically healthy
Remark – the disease in dogs should not be classified or divided into CL, MCL and VL – please correct text
Line 48 – Leishmania spp.
Line 48 – abbreviate Leishmania infantum as L. infantum after its first use
Line 48 – Treatment of dogs for leishmaniosis…
Line 58 – Leishmania
Line 61 – better explain the concepts of as a dietary agent, nutraceutical and phytopharmaceutical agent
Line 64 – write Leishmania in italics – change accordingly throughout the manuscript
Lines 86-88 – “leishmani*” should have been the word – not leishmaniosis or leishmania – the same comment applied to Figure 1
Lines 88-96 – use lowercase initials for Journal, Book chapters, Erratum, Letter, Note, Short review
Line 105 – Total records identified were 178.
Line 113 – the 18 documents
Line 154 – avoid using “n.” – change accordingly throughout the manuscript
Results – use only one decimal place for percentages
Line 164 – Veterinary Parasitology
Line 177 – 10 years
Line 182 ex aequo
Table 2 – unreadable – please revise
Figure 6 – provide complete words or provide captions
Line 332 – display Latin names in italics
The Conclusions section is too long in size – reduce by transferring text into the Discussion section
Comments on the Quality of English LanguageModerate editing of English language required.
Author Response
Dear Reviewer,
Thank you very much for your time and all your comments.
We thank you for your precise and thoughtful comments and constructive criticism, which has led to a better manuscript.
We revised the manuscript in relation to the suggestions and more detailed answers are given below.
The changes made in the manuscript to address comments are written using the
color red.
- In the introduction where you indicate: Leishmania infantum, a flagellate-bear- 29 ing protozoan, is the causative agent, can only Leishmania infantum cause leishmaniasis in dogs? What about Leishmania major and Leishmania tropica? Please expand this part.
- Done.
- In the introduction I don't understand the part that says: Dogs treatment for CanL 48 aims to reduce Leishmania infantum loads by increasing intracellular oxidant compounds to 49 destroy the parasite. However, oxidative species' excess and antioxidants consumption 50 progress towards oxidative stress, resulting in increased, widespread inflammation... what do they mean by that?
- We have reworded the phrase.
- In the introduction, what about treatments with Amphotericin B? pentamidine? paramomycin?
- We have added the treatment with these drugs.
- The discussion does not understand: Various studies have highlighted the potential of 314 immunonutrition in managing inflammatory conditions by mitigating and averting 315 tissue damage caused by oxidative stress. Could you explain this paragraph a little more broadly?
- Done
- In the discussion, explain the following paragraph in more detail: Nutraceutical supplementation could lead 324 to immunomodulation of the Th1 response and a clinical improvement in the animals.
- Done
- Take care of the names of the plants, remember that these are written in italics: Artemisia annua, Plumbago scandens, Physalis angulata, Phyllanthus amarus, 332 Piper aduncum, Peschiera.
- Done
On behalf of the Authors
Prof. Michela Pugliese

Reviewer 3 Report
Comments and Suggestions for Authors
Excellent topic addressed because pharmacological treatments for canine leishmaniasis are discussed and published, however, the nutraceutical area is not discussed. It is important to continue with these lines of research since the host's immune system plays an important role in eliminating the disease. However, I consider that some doubts must be resolved and some corrections made so that the manuscript can be published.
1. In the introduction where you indicate: Leishmania infantum, a flagellate-bear- 29 ing protozoan, is the causative agent, can only Leishmania infantum cause leishmaniasis in dogs? What about Leishmania major and Leishmania tropica? Please expand this part.
2. In the introduction I don't understand the part that says: Dogs treatment for CanL 48 aims to reduce Leishmania infantum loads by increasing intracellular oxidant compounds to 49 destroy the parasite. However, oxidative species' excess and antioxidants consumption 50 progress towards oxidative stress, resulting in increased, widespread inflammation... what do they mean by that?
3. In the introduction, what about treatments with Amphotericin B? pentamidine? paramomycin?
4. The discussion does not understand: Various studies have highlighted the potential of 314 immunonutrition in managing inflammatory conditions by mitigating and averting 315 tissue damage caused by oxidative stress. Could you explain this paragraph a little more broadly?
5. In the discussion, explain the following paragraph in more detail: Nutraceutical supplementation could lead 324 to immunomodulation of the Th1 response and a clinical improvement in the animals.
6. Take care of the names of the plants, remember that these are written in italics: Artemisia annua, Plumbago scandens, Physalis angulata, Phyllanthus amarus, 332 Piper aduncum, Peschiera.
Comments on the Quality of English LanguageMinor editing of English language required.
Author Response
Dear Reviewer,
Thank you very much for your time and all your comments.
We thank you for your precise and thoughtful comments and constructive criticism, which has led to a better
manuscript.
We revised the manuscript in relation to the suggestions and more detailed answers are given below.
The changes made in the manuscript to address comments are written using the
color red.
R. In the introduction where you indicate: Leishmania infantum, a flagellate-bear- 29 ing protozoan, is the causative agent,
can only Leishmania infantum cause leishmaniasis in dogs? What about Leishmania major and Leishmania tropica? Please
expand this part.
A. Done.
R. In the introduction I don't understand the part that says: Dogs treatment for CanL 48 aims to reduce Leishmania
infantum loads by increasing intracellular oxidant compounds to 49 destroy the parasite. However, oxidative species'
excess and antioxidants consumption 50 progress towards oxidative stress, resulting in increased, widespread
inflammation... what do they mean by that?
A. We have reworded the phrase.
R. In the introduction, what about treatments with Amphotericin B? pentamidine? paramomycin?
A. We have added the treatment with these drugs.
R. The discussion does not understand: Various studies have highlighted the potential of 314 immunonutrition in
managing inflammatory conditions by mitigating and averting 315 tissue damage caused by oxidative stress. Could you
explain this paragraph a liVle more broadly?
A. Done
R. In the discussion, explain the following paragraph in more detail: Nutraceutical supplementation could lead 324 to
immunomodulation of the Th1 response and a clinical improvement in the animals.
A. Done
R. Take care of the names of the plants, remember that these are wriVen in italics: Artemisia annua, Plumbago scandens,
Physalis angulata, Phyllanthus amarus, 332 Piper aduncum, Peschiera.
A. Done
On behalf of the Authors
Prof. Michela Pugliese

Round 2
Reviewer 2 Report
Comments and Suggestions for Authors
The authors have addressed all of my comments and accepted all of my suggestions
Comments on the Quality of English LanguageMinor editing of English language required.
Author Response
Dear Reviewer,
Thank you very much for your time and all your comments.
We revised the manuscript concerning the suggestions and the English language. The changes to address comments are in red.
On behalf of the Authors
Prof. Michela Pugliese
